# Cross-border use of health services: An exploratory mixed-methods project at the Mexico-Guatemala border

Ietza Bojorquez[1]*, Marcel Arévalo[2], Ana L. Chávez[2], Rosa N. Gómez-Osorio[2], César Rodríguez-Chavez[3], René Leyva[4], Rachel Gittinger[5], Nirma D. Bustamante[5]

1 Department of Population Studies, El Colegio de la Frontera Norte, Tijuana, Baja California, Mexico, 2 Program on Migration and Poverty, Facultad Latinoamericana de Ciencias Sociales (FLACSO)–Guatemala, Guatemala City, Guatemala, 3 School of Public Health, University of Texas Health Science Center, Houston, Texas, United States of America, 4 Center for Research on Health Systems, Instituto Nacional de Salud Pública, Cuernavaca, Morelos, México, 5 Centers for Disease Control and Prevention, Atlanta, Georgia, United States of America

* ietzabch@colef.mx

## Abstract

Cross-border use of health services has been studied mainly as travel from high- to low- and middle-income countries ("medical tourism"). The movement between low- and middle-income countries has been less studied. The objectives were; 1) to describe the frequency, types of services used, and health needs associated with cross-border utilization of health services at the Mexico-Guatemala border; 2) to explore the drivers of cross-border use among people living in this area. We conducted a mixed-methods study. The quantitative component was a probability survey of border crossers (March to April 2023, analysis sample n = 4,733, weighted n = 74,228). The qualitative component consisted of 28 semi-structured interviews with users and providers of health services living close to the international border (May-June 2023). Descriptive results were obtained separately and triangulated. 3.8% (CI 95% 3.1,4.7) in the sample were crossing the border for the purpose of seeking health care or purchasing medicines, 7.4% (CI 95% 5.9,9.2) had crossed the border in the past year to seek care, and 21.8% (CI 95% 18.8,25.1) to purchase medicines. According to quantitative and qualitative results, those living in Mexico were more likely to cross the border to seek care than those living in Guatemala, independent of country of birth, while crossing to Mexico to buy medicine was more common than crossing to Guatemala for the same reason. Public and private services were accessed in similar proportions, the former mostly for preventive care (vaccination) and the latter for specialized care. Qualitative results showed that the main drivers of cross-border health care use were perceived quality and geographical availability. The main drivers of cross-border buying of medicines were affordability and perceived quality. The use of private services can benefit the local economy. The use of public services for preventive purposes can be an asset for health promotion.

**Data Availability Statement:** All relevant data are within the paper and its Supporting Information files.

**Funding:** This research was conducted with funding from the Cen¬ters for Disease Control and Prevention of the United States of America (Grant: NU50CK000493) through the CDC-Mexico Cooperative Agreement for Surveillance, Epidemiology, and Laboratory Capacity with Fundación México-Estados Unidos para la ciencia (FUMEC). The funders had no role in the study design, data collection and analysis, decision to publish, or preparation of the manuscript.

**Competing interests:** The authors have declared that no competing interests exist.

## Introduction

The use of health services in a foreign country can be driven by multiple reasons, including affordability (reduced prices or entitlement to free care in another country), availability (therapies or drugs that are only available in another country), geographical accessibility (services that are closer in another country), familiarity (knowing the health system in the other country better), and perceived quality (perceiving the system in the other country, or aspects of it, as better) [1, 2]. Different definitions and typologies have been employed to describe this phenomenon, including "cross-border patient movement" [3] and "cross-border patient mobility" [1], where the emphasis is on the demand-side (patients actively seeking care across borders), and "transnational health care" [4] or "cross-border healthcare" [5], where the role of the health system is highlighted. In this article, we employ the term "cross-border use of health services", as an umbrella term to refer to people receiving curative or preventive healthcare, as well as using other health-related services (including laboratory tests, purchasing of medicines, etc.).

Cross-border use of health services is frequent in the border regions of neighboring countries, where these and other services are exchanged as part of the binational dynamics and transnational practices of these areas [1, 3]. This phenomenon has been amply described in the Mexico-United States border region [5], but has also been documented in other contexts [6–10]. Still, the focus of most literature in this area has been on either long-distance travel from high-income countries to low- and middle-income countries (LMICs), or on models for collaboration to allow cross-border health care between high-income countries [1].

The use of health services in LMICs by people living in high-income countries is mostly driven by differentials in income and costs of services, and is commonly referred to as "medical tourism", a potential source of economic development for the recipient countries that mainly involves private services [1, 4]. However, the drivers and consequences of cross-border use between LMICs are less known [3]. In border regions between LMICs, cross-border use can be directed towards public services. An example is the case of patients from Paraguay, Argentina and Peru seeking care in the public Brazilian Unified Health System (SUS), which offers free treatment for conditions not covered in the other countries [8, 9]. Another reason for seeking public health services on the other side of the border can be geographic proximity [2, 9]. Cross-border use for those reasons can challenge the local public health systems of LMICSs. Yet, it can benefit public health in border regions, by facilitating access to preventive actions and life-saving care, as well as public health surveillance. Understanding the characteristics and drivers of cross-border use can help local and national public health systems in their planning, budgeting and policy making [9–11]. Therefore, cross-border use of health services between LMICs is a relevant global health issue, both from the point of view of health systems research, and to inform public health policy.

In this article, we explore cross-border use of health services in the Mexico-Guatemala border region, a zone of intense social and economic exchange between the two countries. The health systems of both countries include social security-associated health care for those with formal employment, private providers ranging from low- or no-cost pharmacy-adjacent doctor's offices to expensive top-quality specialized care, and governmental health care facilities, basically open to all persons, but in practice covering mainly those of lower income (who can't afford other types of care) or living in rural areas [12, 13]. However, the resources available are very different: Guatemala's public health expenditure as percentage of its GDP was 2.3% in 2021, while Mexico's was 3%. Guatemala had 4.4 hospital beds per 10,000 inhabitants in 2017, while Mexico had 9.8 in 2018. The number of doctors per 10,000 inhabitants in 2020 was, respectively, 12.8 and 24.4 [14]. These differentials could impulse border crossings for health care, as in some of the cases mentioned above [2, 8, 9]

Our specific aims for this project were: 1) to describe the frequency, types of services used, and health needs associated with cross-border utilization of health services at the Mexico-Guatemala border; 2) to explore the drivers of cross-border use in this population.

## Methods

A mixed-methods project was conducted from March to July 2023. We followed a concurrent triangulation design [15], in which qualitative (QUAL) and quantitative (QUAN) data were collected independently, analyzed separately, and triangulated for the final analysis.

The QUAN component consisted of a survey in three border towns in Guatemala (La Mesilla, El Carmen and Tecún Umán) with the highest numbers of registered border crossings of the Mexico-Guatemala border. For the QUAL component, we interviewed informants living or working in those same towns as well as in Ciudad Cuauhtémoc and Comitán in Mexico (cities that are close or adjacent to the border).

### QUAN component

We conducted a survey from March 15 to April 15, 2023, of people entering Guatemala from Mexico. We followed a time-venue sampling design, in which combinations of day/hours and sampling points were selected with probability proportional to size, using a sampling frame developed from previous surveys and from visits to the field to update it. Within each time-venue combination, a systematic sampling procedure was followed to approach potential respondents and screen them for eligibility. This sample selection method was developed for the Surveys of Migration in Mexico's Borders (EMIF) and has been employed by the authors in the past [10, 11]. It is designed to obtain representative estimations for the population of crossings in a given period of time. A more detailed description can be read in https://www.colef.mx/emif/diseno.html. The eligibility criteria were: 1) $\geq$ 18 years old; 2) having crossed the border within the past 12 hours.

For the purposes of comparison, we classified the QUAN sample in four groups: Guatemalans living in Guatemala (GG) (n = 3,722), Guatemalans living in Mexico (GM) (n = 167), Mexicans living in Mexico (MM) (n = 685) and other (O) (n = 159). The last category included people born in other countries (n = 141), Guatemalans living in other countries (n = 12) and Mexicans living in Guatemala (n = 6), collapsed together because of the small sample sizes for each combination of country of birth- country of residence.

We defined cross-border utilization of health services as the use of preventive or curative health care services, or purchasing of medicines, in a country other than the one the person lived in. As indicators, we employed the responses to the questions: "What is/was the main purpose of your visit to Guatemala/Mexico today?", "In which country do you and your family usually seek care?", "In the past year, did you ever travel to a country other than the one you live in, to seek care/purchase medicines?", "In which country did you receive care for your more recent health need?" (for those reporting having had a health need in the past two weeks), "In which country were you vaccinated for COVID-19"? and "In which country do you and your family usually get vaccinated?"

Interviewers approached 11,185 persons, of whom 4,757 were eligible and agreed to participate. After applying weights, the sample represents 74,673 border crossings. The analysis in this article is based on the 4,733 participants (weighted n = 74,228) who gave information of their country of birth and residence. All data reported in the results is weighted unless otherwise indicated and considers the sample design (strata and clusters). The analysis was conducted with Stata v.17 [16].

## QUAL component

From June 6 to June 29, 2023, we interviewed 14 health care providers and 14 users of health services, selected by purposive sampling through visits to health care facilities and pharmacies, contacts of the research team, and snowball sampling from the first interviewees. At the time of this research, the area was undergoing severe criminal violence, so some interviews were conducted online for security reasons. The interviews lasted from 15 to 40 minutes. The characteristics of participants appear in Table 1.

The semi-structured interview guide asked about informants' knowledge or personal experience along four dimensions: 1) cross-border use of health services in the border region; 2) the main reasons for going across the border for care (including vaccination) or to purchase medicines; 3) the health needs that more frequently motivated cross-border use; the types of services used; and 4) the characteristics of persons who were more likely to engage in cross-border use. Four of the authors (IB, MA, ALC, RGO) conducted the interviews in Spanish. All interviewers are native Spanish speakers and investigators with post-graduate studies who are familiar with the region and the population.

The interviews were audio-recorded and transcribed, and the four authors heard the transcripts to familiarize themselves with the ones they hadn't conducted. Then, following a deductive analytic strategy, they prepared a matrix with cases (participants) on the rows and the dimensions described in the previous paragraph in the columns, summarizing what was said by participants into the matrix's cells. Afterwards, they discussed the results and summarized the results for each dimension.

**Table 1. Characteristics of participants in the qualitative component (n = 28).**

| Characteristic | Number of participants |
|---|---:|
| Service provider (n = 14) | |
| Gender | |
| Male | 8 |
| Female | 6 |
| Age group | |
| 18–44 | 9 |
| 45–64 | 3 |
| 65+ | 2 |
| Country of residence | |
| Mexico | 1 |
| Guatemala | 13 |
| Service user (n = 14) | |
| Gender | |
| Male | 3 |
| Female | 11 |
| Age group | |
| 18–44 | 6 |
| 45–64 | 7 |
| 65+ | 1 |
| Country of residence | |
| Mexico | 4 |
| Guatemala | 10 |

## Mixed methods analysis

After analyzing the QUAN and QUAL results separately, we developed a second matrix with the four dimensions in the columns, and the component (QUAN or QUAL) in the rows. We mapped the QUAN and QUAL results into the matrix and used it as the tool for extracting the main conclusions. In doing this, we moved iteratively between QUAL and QUAN data, with results from one component inspiring new analyses of the other until new analyses led to no new conclusions.

## Ethics statement

Eligible participants read or were read an informed consent in Spanish, and provided their consent to participate in the survey or the interview. The audio recordings were assigned an alphanumeric code to ensure the interviewee's anonymity. Personal information was accessed only by investigators. Since this was considered a minimal risk research, the IRB authorized verbal consent from participants to further preserve confidentiality. The protocol was reviewed and approved by the IRB of El Colegio de la Frontera Norte (no. 079_230821). This activity was reviewed by CDC and was conducted consistent with applicable federal law and CDC policy. (See e.g., 45 C.F.R. part 46, 21 C.F.R. part 56; 42 U.S.C. §241(d); 5 U.S.C. §552a; 44 U.S.C. §3501 et seq.).

## Inclusivity in global research

Additional information regarding the ethical, cultural, and scientific considerations specific to inclusivity in global research is included in the Supporting Information (S1 Text).

## Disclaimer

The findings and conclusions in this report are those of the authors and do not necessarily represent the official position of the Centers for Disease Control and Prevention.

## Results

### Quantitative results

The majority (76.3%) of border crossers in the survey were GG, followed by MM (16.1%) (Table 2). About half (47.7%) were women. The mean age was 36.6 years, with MM slightly older than GG and O. Close to a third of respondents (29.6%) considered themselves indigenous, with the highest percentage among GG. The educational level was low on average, especially among GG, of whom 38.0% had less than six years of education. The main reasons for the crossing at the time of the survey were for work or business and visiting friends or family. Of the whole sample, 3.8% reported health-related reasons for crossing, with the highest percentage among MM (9.4%).

As for the country where participants and/or their families usually sought care (Table 3), most of them reported it was the country they lived in. However, over a fifth (22.9%) of GM used services in Guatemala, as did 8.1% of MM, and 2.6% of GG used services in Mexico. Moreover, 7.4% of the whole sample had travelled to a different country in the past year for health care. Crossing borders to purchase medicines was even more common: 24.0% of GG had visited Mexico for that reason in the past year, and 14.4% of GM and 14.0% of MM had visited Guatemala because of it.

When asked about recent health needs, 21.7% of the sample reported having had a health need in the two weeks prior to the interview (Table 3). For those who received care for that need, among GG similar percentages had received care in Guatemala and Mexico (48.7% vs

**Table 2. Characteristics of participants in the quantitative survey.**

| Variable | Country of birth/residence | | | | |
|---|---|---|---|---|---|
| | Guatemalans living in Guatemala (n = 3,722)[a] | Guatemalans living in Mexico (n = 167)[a] | Mexicans living in Mexico (n = 685)[a] | Other (n = 159)[a] | Total (n = 4,733)[a] |
| Estimated % of the population (CI) [b] | 76.3 (73.5,78.9) | 4.2 (3.4,5.3) | 16.1 (14.0,18.4) | 3.4 (2.7 (4.3) | 100 |
| Female (%, CI) | 46.3 (44.3,48.3) | 61.6 (51.5,70.8) | 50.4 (46.1,54.6) | 50.1 (39.1,61.0) | 47.7 (45.9,49.5) |
| Age (mean, CI) | 36.1 (35.6,36.7) | 38.7 (36.3,41.2) | 39.1 (37.5,40.7) | 33.6 (31.3,35.9) | 36.6 (36.1,37.2) |
| Ethnicity (%, CI) | | | | | |
| Indigenous | 31.8 (29.1,34.7) | 30.3 (23.6,37.9) | 25.0 (21.3,29.0) | 0.5 (0.1,4.2) | 29.6 (27.3,32.0) |
| Afrodescendant | 0.1 (0.0,0.3) | 0 | 0.2(0.0,1.8) | 2.6 (0.6,10.3) | 0.2 (0.1,0.5) |
| Both | 0.2 (0.1,0.5) | 0.3(0.0,2.0) | 0 | 0 | 0.2 (0.1,0.4) |
| None of the above | 67.9 (65.1,70.6) | 69.5 (61.8,76.2) | 74.8 (70.4,78.7) | 96.9 (90.0,99.1) | 70.1 (67.6,72.4) |
| Years of education (%, CI) | | | | | |
| 0–5 | 38.0 (34.7,41.4) | 28.6 (19.0,40.6) | 14.9 (10.6,20.7) | 15.2 (9.0,24.6) | 33.1 (30.0,36.3) |
| 6–9 | 49.2 (46.2,52.1) | 51.8 (39.7,63.7) | 55.4 (49.0,61.6) | 45.3 (37.7,53.3) | 50.2 (47.1,53.2) |
| 10–12 | 11.7 (10.0,13.6) | 16.8 (9.4,28.2) | 21.7 (18.4,25.3) | 20.3 (12.3,31.7) | 13.8 (12.0,15.9) |
| 13+ | 1.2 (0.9,1.6) | 2.8 (1.1,7.1) | 8.0 (5.6,11.4) | 19.2 (12.5,28.3) | 3.0 (2.4,3.6) |
| Main reason for visiting Mexico or Guatemala[c] (%, CI) | | | | | |
| Working/business | 77.8 (75.0,80.3) | 16.9 (9.4,28.5) | 38.4 (33.1,43.9) | 64.7 (40.4,83.2) | 68.7 (65.8,71.4) |
| Visiting family or friends | 17.3 (15.1,19.9) | 74.3 (59.0,85.4) | 47.0 (41.9,52.1) | 26.3 (9.9,53.7) | 24.7 (22.0,27.5) |
| Health-related | 2.7 (2.0,3.5) | 5.5 (2.2,13.5) | 9.4 (6.7,12.9) | 0 | 3.8 (3.1,4.7) |
| Other | 2.2 (1.6,3.1) | 3.2 (1.2,8.2) | 5.3 (3.1,8.9) | 9.0 (3.1,23.6) | 2.8 (2.2,3.7) |

[a] Unweighted n. All other data in Table are weighted.

[b] CI = confidence interval 95%.

[c]Visiting Mexico if living in Guatemala, visiting Guatemala if living in Mexico

51.0%, respectively), while most GM (85.4%) received services in Mexico. MM had mainly received care in Mexico for their recent health need (66.6%), but 33.4% had received care in Guatemala for it.

Further analysis of the type of needs and services involved in cross-border health care use was limited by small sample sizes for some groups. However, for GG the more frequent reasons reported were injuries or accidents (40.5%, CI95% 30.0,52.0), gastrointestinal illness (18.1%, CI95% 12.0,26.3), and maternal care (12.9%, CI95% 7.9,20.4) (not shown in Tables). Only GG reported cross-border use of maternity services. Other health reasons reported with less frequency included chronic illnesses (6 cases among GG), dental care (11 cases among GG) and ophthalmological care (6 cases among MM) (not shown in Table).

Small sample sizes in some cells likewise limits the analysis of the type of health facility cross-border users went to, but among GG (the group with the biggest sample size), close to half of those who sought care in Mexico in the past two weeks had been seen in public services

**Table 3. Characterization of cross-border health services use.**

| Variable | Guatemalans living in Guatemala (n = 3,722)[a] | Guatemalans living in Mexico (n = 167)[a] | Mexicans living in Mexico (n = 685)[a] | Other (n = 159)[a] | Total (n = 4,733)[a] |
|---|---|---|---|---|---|
| **Country of birth/residence** | | | | | |
| Country where participant/family usually seek health care (%, CI)[b] | | | | | |
| Guatemala | 97.3 (96.2,98.1) | 22.9 (16.8,30.2) | 8.1 (6.0,10.9) | 17.7 (9.5,30.8) | 77.6 (75.2,79.8) |
| Mexico | 2.6 (1.8,3.7) | 77.2 (69.8,83.2) | 91.9 (89.1,94.0) | 9.1 (5.3,15.4) | 19.9 (17.9,22.1) |
| Other | 0.1 (0.0,0.5) | 0 | 0 | 73.1 (61.3,82.4) | 2.6 (2.0,3.3) |
| Crossed the border in the past year to. . . (%, CI) | | | | | |
| seek care | 6.9 (5.4,8.9) | 7.7 (2.5,21.7) | 9.9 (7.1,13.7) | 4.5 (2.1,9.3) | 7.4 (5.9,9.2) |
| buy medicines | 24.0 (20.4,27.9) | 14.4 (8.2,24.1) | 14.0 (10.5,18.4) | 18.5 (11.3,28.9) | 21.8 (18.8,25.1) |
| Had a health care need in the past two weeks | 23.7 (20.4,27.3) | 6.5 (3.5,12.0) | 16.5 (11.9,22.4) | 20.7 (13.0,31.3) | 21.7 (18.7,25.1) |
| Country where participant received care[c] | | | | | |
| Guatemala | 48.7 (40.0,57.5) | 14.6 (3.3,46.4) | 33.4 (18.3,52.9) | 31.1 (10.8,62.9) | 45.3 (38.6,52.3) |
| Mexico | 51.0 (42.2,59.8) | 85.4 (53.6,96.7) | 66.6 (47.2,81.7) | 65.5 (35.4,86.7) | 54.3 (47.3,6.1) |
| Other | 0.3 (0.1,1.1) | 0 | 0 | 3.4 (0.4,22.7) | 0.4 (0.1,1.2) |
| Vaccinated against COVID-19 | 74.8 (72.5,77.0) | 74.6 (65.1,82.2) | 86.8 (86.5,90.2) | 79.3 (71.7,85.3) | 76.8 (74.7,78.8) |
| Country of vaccination against COVID-19[d] | | | | | |
| Only Guatemala | 92.4 (90.3,94.1) | 11.9 (5.8,23.1) | 2.3 (1.5,3.6) | 10.8 (5.4,20.3) | 70.7 (67.7,73.6) |
| Only Mexico | 6.8 (5.2,8.9) | 83.8 (71.0,91.6) | 96.8 (95.2,97.9) | 11.9 (5.6,23.6) | 25.9 (23.1,28.9) |
| Guatemala and Mexico | 0.3 (0.1,0.8) | 3.4 (1.2,9.5) | 0.3 (0.1,1.2) | 0.3 (0.0,2.3) | 0.4 (0.2,0.8) |
| Other | 0.5 (0.1,1.5) | 0.8 (0.1,6.1) | 0.6 (0.1,2.2) | 77.0 (62.8,87.0) | 3.0 (2.2,4.0) |
| Country where participant and family usually get vaccines | | | | | |
| Guatemala | 98.1 (97.3,98.7) | 21.7 (15.2,30.0) | 3.6 (2.0,6.2) | 16.9 (9.3,28.8) | 77.6 (75.1,79.9) |
| Mexico | 1.9 (1.3,2.7) | 78.3 (70.0,84.8) | 96.4 (93.8,98.0) | 8.2 (4.5,14.5) | 19.8 (17.7,22.1) |
| Other | 0 | 0 | 0 | 74.9 (64.3,83.2) | 2.6 (2.0,3.4) |
| They don't usually get vaccinated | 0[5] | 0 | 0 | 0 | 0[e] |

[a] Unweighted n. All other data in Table are weighted.

[b] CI = confidence interval 95%.

[c] Percentages among those who had a health need in the past two weeks and received health care for it.

[d] Percentages over those vaccinated against COVID-19.

[e] Only one participant responded with this option.

(either hospitals or clinics) (48.6%, CI95% 36.4,60.9), with the rest distributed between pharmacy-adjacent offices (31.1%, CI95% 22.5,41.3) and other private services (hospitals, clinics, doctors' offices) (20.3%, CI95% 13.2,29.9) (not shown in Table).

With regard to vaccination, of GG vaccinated against COVID-19, 92.4% had received the vaccine in Guatemala, and 83.8% of GM and 96.8% of MM had received it in Mexico (Table 3). Most GG (98.1%) reported that they and their families were usually vaccinated in Guatemala, while 78.3% of GM and 96.4% of MM were usually vaccinated in Mexico.

## Qualitative results

According to interviewees, the main health-related reasons for people to cross the border was purchasing medicines, but cross-border health care use was also described, mainly for specialist services, maternity services (women living in Guatemala going to Mexico for delivery), and dentistry, as exemplified in the following quotes:

> "Regarding medicines, I have had the experience of buying medicines for family members or for myself, over there, in Tapachula [Mexico]. There are people that knows that I go there and bring back medicines, and they ask me to bring them some. They give me the money or I tell them how much they will cost, and they give me the cost, or they pay my travel expenses, for me to go there and bring the medicines. And yes, I have needed to go often for medicines for people who has diabetes, to bring insulin pens, or medicines for persons who have seizures, or for constipation, they only give me their prescriptions and I go and buy them there. And it's good for them because of the exchange rate, it's less expensive for them. Here in Guatemala the prices are a little bit higher and therefore it's better for them to buy in the Mexican side." (U-M-0002-P, user, Guatemala)

> "[There are] people who want to deliver in the other side, in the Mexican side. Maternity. Gynecology services. *Interviewer*: Why, is the care better over there? *Participant*: First, because Comitán [in the Mexican side] is closer. The highway is better. Not like in Huehuetenango [in the Guatemalan side], where the highway is quite bad. So, they prefer to go to Mexico." (LM02, provider, Guatemala)

Reports of COVID-19 vaccination in the qualitative component were supported by quantitative results, which showed that most participants received the COVID-19 vaccine in the country where they lived, with the exception of Guatemalans living in Mexico–who received the COVID-19 vaccine in Mexico. No clear trends were appreciated among other vaccines: there were examples of tetanus vaccine provided to Mexicans in Guatemala, HPV vaccine to Guatemalans in Mexico, and children completing their vaccination scheme on both sides of the border. An example is the following quote:

> "*Interviewer*: Do they come [from Mexico] to get vaccinated? *Participant*: Yes, they can't get the tetanus vaccine in Mexico and they come here to get vaccinated. Sometimes there's no vaccine here either and then people have to buy it in pharmacies [. . .] *Interviewer*: Is it common, that they ask for the tetanus vaccines? *Participant*: Yes, I've heard that a lot. It's frequent here, they are from *aldeas* [rural communities] and they fall down or get cut in the corrals or have some type of accident, and they come because there's no vaccine in the public clinics there" (P-EC-0001-P. Provider. Guatemala)

As for the type of services, participants mentioned cross-border use of both private and public services. The private services included doctors (general practitioners and specialists),

nurses, dentists, psychologists, and physical therapists, as well as pharmacies and laboratories, both in Mexico and Guatemala. The public services mentioned were mainly preventive (vaccinations, preventive campaigns). One exception was the public maternity hospital in Comitán [Mexico] referred to above, that was mentioned by several interviewees as a place where it was common for women from Guatemala to seek care.

Affordability was a driver of cross-border use of health services frequently mentioned for buying medicines in Mexico. As described in the quote above, the exchange rate of quetzal to peso explained this practice, which according to interviewees was facilitated by the lack of customs control for people entering Guatemala from Mexico.

In contrast, crossing from Mexico into Guatemala to buy medicines was driven mostly by preference. Some medicines and other health-related products were only sold there, and they were sought after by people living in Mexico who were familiar with them since childhood. One participant mentioned that this was probably a vestige of past decades, when the Mexican towns near the border were isolated from Mexico, and many of their manufactured goods came from Guatemala. Another reason for preferring medicines bought in Guatemala was that this historical exchange has familiarized people in Mexico with the brand names used in Guatemala, so brand trust was also at play. Others confirmed that there is a tradition of those living in the Mexican side buying products in Guatemala, and of Guatemalan street vendors coming to Mexican towns to sell healing herbs, vitamins and other dietary supplements, and medicines.

Cross-border use of health care from Guatemala to Mexico, on the other hand, was driven mostly by availability. Most Guatemalan communities in the border area are small, and far from the major urban centers in that country, but with good roads connecting them to the border, so that some services (as with maternity care) are easier to access in the Mexican side. This motivated some participants to go to Mexico for specialized care, while staying in Guatemala for general medical consultations. The tariffs of Mexican doctors were considered more expensive by some interviewees, and less expensive by others, so affordability was not the main reason for cross-border use of health care in Mexico. The following quote exemplifies this:

"*Interviewer*: Have you ever used medical services for yourself or a family member in the Mexican side? *Participant*: Not for myself, but for a family member. My brother [needed] private medical care in Tapachula [Mexico], because of kidney problems, kidney stones, he needed a laser surgery, in a private clinic. It was an emergency since in the Malacatán [Guatemala] region there is no kidney specialist, not till Quetzaltenango. Then, the emergency was pressing, then, it was necessary to go to the Mexican side, to Tapachula, where it's a little bit more. . . [doesn't finish the phrase]" (U-M-0002-P. User. Guatemala).

People living in Mexico crossed to Guatemala for specialized care too. Many people mentioned they would cross into Malacatán, a town of about 100,000 inhabitants 15km from the international border, to see an ophthalmologist. Another reason people would cross was to receive HIV care in the same town. In these cases, a combination of perceived quality and availability was the main driver. Another reason for cross-border use of HIV care was avoiding stigma since people preferred to seek specialized care in a town where they were unknown. Perceived quality was the most relevant driver for people making longer trips to see specialists in Guatemala City, recommended to them by family or friends.

Another driver of cross-border health use was what Rodriguez-Chavez, Larrea-Schiavon (10) call "circumstantial" use, i.e. people visiting the neighboring country for other reasons, and then having an acute health need while there, or else taking advantage of the trip to seek care or purchase medicines as a second thought. This was mentioned by many participants

who highlighted this as a most common cause of cross-border health use. They mentioned, for example, people crossing to Guatemala to buy or sell something, and getting vitamins or other medicines while at the market. They also described people who lived in Mexico and crossed every day to work in the fields in the Guatemalan side, and how they took their children to be vaccinated in the public clinics in Guatemala, since the clinics in the Mexican side of the border would be closed by the time they went back there.

As for vaccination, most participants mentioned that the COVID-19 vaccine had been available earlier on the Mexican side, so that many people crossed from Guatemala to get vaccinated there. Later, the Pfizer vaccine was only available in Mexico, and some people living in Guatemala went across the border because of the perceived quality of that vaccine. Thus, a combination of availability and perceived quality was at play in the decision of where to seek COVID-19 vaccination.

In contrast, cross-border vaccination for other diseases was driven by availability and occurred in both directions. A participant mentioned that sometimes there were preventive campaigns in Guatemala, and people came from the Mexican side to benefit from them, and that some vaccines were only available in Guatemala. But there were also mentions to vaccines being more readily available in Mexico.

Few participants considered that cross-border use could be a challenge for the public health system. However, one participant who worked in that system in Guatemala said they sometimes had trouble adjusting their stocks of vaccines to the demand, since the number of people living in Mexico who sought vaccination in her clinic varied substantially ~~a lot~~. The informant emphasized that vaccination and other care was provided without asking nationality or country of residence, a fact that was mentioned by other participants in reference to both the Guatemalan and Mexican public health system. This was related to the observation by one participant that there were no real controls of border crossing, and the porous borders facilitated cross-border use of services.

When asked what types of persons or social groups where more likely to engage in cross-border use of health services, participants underscored the circumstantial use mentioned above, when medicines were purchased, and services sought while in the course of other cross-border activities. Thus, the ones more likely to make cross-border use of health services would be those already engaged in the transnational dynamics of the area, including members of transnational families, transnational workers, or those with businesses in the neighboring country.

On the other hand, while affordability was mentioned as a driver of cross-border purchase of medicines, it still implied some degree of economic capacity; therefore, people with higher income would be the ones more likely to engage in it. The same would apply to those able to choose where to seek care for their needs, such as maternity services in Mexico, or doctors in Guatemala City. At the same time, agricultural workers from Mexico were described as making use of public services in Guatemala, so it seems different social groups accessed different types of services: people of higher income would purchase medicines and seek private or public care according to preference, while those with more restricted means would make circumstantial use of public health care while working in the neighboring country.

## Mixed methods conclusions

Both the QUAN and QUAL components showed that the most frequent type of cross-border use of health services was purchasing medicines, especially among GG. However, according to the QUAN component, health-related reasons were rarely the main reason for border crossing, and few survey respondents reported cross-border health care as their usual source of

care. On the other hand, 51.0% of GG and 33.4% of MM with a recent health need had used services across the border. Combined with the QUAL data, this strengthens the conclusion of a circumstantial use of cross-border health care, which may not be perceived as "usual care" by people but still occurs with a certain frequency.

The main needs for which cross-border care was sought, according to the QUAN component, were injuries/accidents, respiratory infections, gastrointestinal problems, and maternal care. In contrast, QUAL data highlighted specialized care. It is possible that the difference is related to the salience of each type of condition in the interviewees' perception. It is also interesting to notice that the more frequent health needs mentioned in the "other" category in the QUAN survey included the same conditions that the QUAL interviewees described (diabetes, dental and ophthalmological care), which some of the quote examples in this manuscript showcase.

According to QUAN results, public services were used by about half (48.6%) of the GG that received care in Mexico for a recent health need. QUAN data also showed that cross-border vaccination was infrequent, but it was more common in the case of COVID-19. This coincides with the QUAL data describing how GG went across the border for this vaccine, while for other vaccines the movement could also happen from Mexico to Guatemala, depending on the local availability. At least one informant considered that this represented a challenge for the planning of vaccine stocks in the local public services.

As for the drivers of cross-border use, the QUAN component showed that MM were the ones that more frequently had health-related reasons as their main reason for crossing, while a higher percentage of GG had crossed in the past year with the purpose of buying medicines, and GM were the ones more likely to report cross-border health care as their usual source of care. The QUAL data provides richer information in this regard. Combining the two types of information, the main driver of cross-border use of health care (including vaccination) in this population is a combination of availability and perceived quality, while cross-border purchasing of medicines is driven mainly by affordability.

## Discussion

Our first aim in this mixed methods project was to characterize the cross-border use of health services. We found that, even though few people crossed the border solely for health-related reasons, many border crossers had received care in the neighboring country. Cross-border purchasing of medicines was even more frequent, especially among GG. Cross-border care was sought for all types of health needs, including accidents, acute and chronic illnesses, maternal care, and vaccination. As for the types of services used, similar percentages of GG had resorted to private and public health care in Mexico for a recent health need. While our QUAN data doesn't allow us to estimate the frequency of cross-border use of private versus public health care in Guatemala, according to the QUAL data the distribution may be similar.

Regarding our second aim, the drivers of cross-border use in this population were a combination of affordability (in the case of medicines bought in Mexico), availability (of some medicines in Guatemala, and of specialized services and vaccines in both sides of the border), familiarity (with medicines sold in Guatemala), and perceived quality (of medicines, health providers or vaccines). From our results, upstream level drivers can also be inferred, and cross-border use of health services in this population can be explained by the historical social, cultural, and economic binational integration of the region. Thus, the common practice of going across the border for work or commerce, and the existence of transnational families, facilitated the circumstantial use of services, which could explain why crossing solely for health-related purposes was infrequent, while circumstantial cross-border use of health

services was more common. Culturally, people in the region are familiar with the medicines sold in Guatemala, which motivates cross-border purchase by people living in Mexico. Another upstream driver of cross-border use was the urban development and the network of roads, which made services in Mexico more available for some of those living in the Guatemalan border region. Finally, multiple political and economic reasons made the monetary exchange rate favor those with quetzales and explained the frequent practice of purchasing medicines in the Mexican side by those living in Guatemala.

Our results contrast with the case of Brazil, where the existence of a universal system providing health care free at the point of delivery ~~at no cost~~ and regardless of migration status attracts cross-border use [8, 9]. The cost of care was not a major driver in our project, and those using the public system were motivated mainly by availability (of vaccines) and perceived quality (the case of the Mexican public maternity hospital). Instead, they replicate what we reported in a previous body of work, in which accidents were the most frequent health problem among GG, 65.4% of whom has received care in Mexico for their last health-related need [10]. Similar to our results, availability was the main driver of cross-border use of health care in a study in the Dominican Republic–Haiti border area [2]. Also similar are the results of a study in the Laos-Thailand border, reporting that social networks were a main driver of patients' decision to use health services across the border [7].

A strength of our analysis was that it used a probability sample design, representative of the population of border crossings at the time of the study. However, this resulted in smaller sample sizes for people living in Mexico (as compared to GG), so that some estimates could not be computed for them. Another limitation was that the QUAL interviews included a small number of people living in the Mexican side, so the results mostly represent the view of those living in Guatemala. In the future, we aim to recruit a more diverse sample, so we can verify if our conclusions stand. Lastly, our QUAN survey was only implemented in Spanish. Therefore, linguistically diverse populations were not represented.

To conclude, cross-border use of health services is a common feature of life in the Guatemala-Mexico border. Even though our data are limited in this regard, it seems that cross-border use is similarly directed towards public and private health care, but the most common use is buying medicine from private providers, thus representing an economical opportunity for the Mexican side. Cross-border use of public health care was mainly for preventative services, and this can be a regional asset for the health of the border population. Both countries may want to consider this in their public health policy planning, making use of this asset to strengthen their capacity to promote the health of this binational community.

## Supporting information

**S1 Text. "Authors' responses to Inclusivity in Global Research Questionnaire".** (DOCX)

**S1 Data. "Minimum data set to replicate findings".** (DTA)

## Author Contributions

**Conceptualization:** Ietza Bojorquez.

**Data curation:** Ietza Bojorquez, Marcel Arévalo, Ana L. Chávez, Rosa N. Gómez-Osorio.

**Formal analysis:** Ietza Bojorquez, Marcel Arévalo, Ana L. Chávez, Rosa N. Gómez-Osorio.

**Funding acquisition:** Ietza Bojorquez, Nirma D. Bustamante.

**Investigation:** Ietza Bojorquez, Marcel Arévalo, Ana L. Chávez, Rosa N. Gómez-Osorio.

**Methodology:** Ietza Bojorquez, Marcel Arévalo, Ana L. Chávez, Rosa N. Gómez-Osorio, Nirma D. Bustamante.

**Project administration:** Ietza Bojorquez, Marcel Arévalo, Nirma D. Bustamante.

**Supervision:** Ietza Bojorquez, Marcel Arévalo.

**Writing – original draft:** Ietza Bojorquez.

**Writing – review & editing:** Ietza Bojorquez, Marcel Arévalo, Ana L. Chávez, Rosa N. Gómez-Osorio, César Rodríguez-Chavez, René Leyva, Rachel Gittinger, Nirma D. Bustamante.

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
