## [Decision Letter · Decision Letter 0]

8 Aug 2024

PGPH-D-24-01431

Cross-border use of health services: An exploratory mixed-methods project at the Mexico-Guatemala border

Dear Dr. Bojorquez,

Thank you for submitting your manuscript to PLOS Global Public Health. After careful consideration, we feel that it has merit but does not fully meet PLOS Global Public Health’s publication criteria as it currently stands. Therefore, we invite you to submit a revised version of the manuscript that addresses the points raised during the review process.

One of the reviewers have indicated some minor revisions that I am inviting you to address.

We look forward to receiving your revised manuscript.

Kind regards,

Ferdinand C Mukumbang, PhD

Academic Editor

Journal Requirements:

Additional Editor Comments (if provided):

Reviewers' comments:

Reviewer's Responses to Questions

**Comments to the Author**

1. Does this manuscript meet PLOS Global Public Health’s publication criteria? Is the manuscript technically sound, and do the data support the conclusions? The manuscript must describe methodologically and ethically rigorous research with conclusions that are appropriately drawn based on the data presented.

Reviewer #1: Yes

2. Has the statistical analysis been performed appropriately and rigorously?

Reviewer #1: I don't know

3. Have the authors made all data underlying the findings in their manuscript fully available (please refer to the Data Availability Statement at the start of the manuscript PDF file)?

Reviewer #1: No

4. Is the manuscript presented in an intelligible fashion and written in standard English?

Reviewer #1: Yes

5. Review Comments to the Author

Reviewer #1: This is undoubtedly an interesting paper that demonstrates the cross-border features of the use of health services in two countries belonging to the category of low and middle income countries (LMIC).

However, I lacked a bit of data triangulation in this work. Are there any additional or related data that could support the authors' conclusions given the sample size?

In addition, the work would have been richer if the authors had briefly described the characteristics of the health care system of both countries. It also affects the search for services in a neighboring country.

Furthermore, I think it would be useful and also interesting to illustrate the example of other countries to compare the findings and trends. The work just only briefly mentions the example of Brazil.

I am asked about the availability of all data, and the authors indicate the following:

"The quantitative data used for this article will be made publicly available in the first author's institutional repository. The qualitative data contains sensitive information and therefore cannot be shared." However, I did not see any link or redirect to access the quantitative data. And looking for the repository of the first author's institution takes time. I may have missed it somewhere, so it would be helpful if this information was somehow presented in a way that would not take time to search for it further.

6. PLOS authors have the option to publish the peer review history of their article (what does this mean?). If published, this will include your full peer review and any attached files.

**Do you want your identity to be public for this peer review?** For information about this choice, including consent withdrawal, please see our Privacy Policy.

Reviewer #1: No

---

## [Editor Report · Decision Letter 1]

5 Sep 2024

Cross-border use of health services: An exploratory mixed-methods project at the Mexico-Guatemala border

PGPH-D-24-01431R1

Dear Dr. Bojorquez,

We are pleased to inform you that your manuscript 'Cross-border use of health services: An exploratory mixed-methods project at the Mexico-Guatemala border' has been provisionally accepted for publication in PLOS Global Public Health.

Best regards,

Ferdinand C Mukumbang, PhD

Academic Editor